# VA-DepthNet: A Variational Approach to Single Image Depth Prediction

**Ce Liu[1]   Suryansh Kumar[1]\*  Shuhang Gu[2]   Radu Timofte[1,3]   Luc Van Gool[1,4]**
[1]CVL ETH Zürich  [2]UESTC China   [3]University of Würzburg   [4]KU Leuven
{ce.liu, sukumar, vangool}@vision.ee.ethz.ch
 shuhanggu@uestc.edu.cn, radu.timofte@uni-wuerzburg.de

## Abstract

We introduce VA-DepthNet, a simple, effective, and accurate deep neural network approach for the single-image depth prediction (SIDP) problem. The proposed approach advocates using classical first-order variational constraints for this problem. While state-of-the-art deep neural network methods for SIDP learn the scene depth from images in a supervised setting, they often overlook the invaluable invariances and priors in the rigid scene space, such as the regularity of the scene. The paper's main contribution is to reveal the benefit of classical and well-founded variational constraints in the neural network design for the SIDP task. It is shown that imposing first-order variational constraints in the scene space together with popular encoder-decoder-based network architecture design provides excellent results for the supervised SIDP task. The imposed first-order variational constraint makes the network aware of the depth gradient in the scene space, i.e., regularity. The paper demonstrates the usefulness of the proposed approach via extensive evaluation and ablation analysis over several benchmark datasets, such as KITTI, NYU Depth V2, and SUN RGB-D. The VA-DepthNet at test time shows considerable improvements in depth prediction accuracy compared to the prior art and is accurate also at high-frequency regions in the scene space. At the time of writing this paper, our method—labeled as VA-DepthNet, when tested on the KITTI depth-prediction evaluation set benchmarks, shows state-of-the-art results, and is the top-performing published approach[1,2].

## 1 Introduction

Over the last decade, neural networks have introduced a new prospect for the 3D computer vision field. It has led to significant progress on many long-standing problems in this field, such as multi-view stereo (Huang et al., 2018; Kaya et al., 2022), visual simultaneous localization and mapping (Teed & Deng, 2021), novel view synthesis (Mildenhall et al., 2021), etc. Among several 3D vision problems, one of the challenging, if not impossible, to solve is the single-image depth prediction (SIDP) problem. SIDP is indeed ill-posed—in a strict geometric sense, presenting an extraordinary challenge to solve this inverse problem reliably. Moreover, since we do not have access to multi-view images, it is hard to constrain this problem via well-known geometric constraints (Longuet-Higgins, 1981; Nistér, 2004; Furukawa & Ponce, 2009; Kumar et al., 2019; 2017). Accordingly, the SIDP problem generally boils down to an ambitious fitting problem, to which deep learning provides a suitable way to predict an acceptable solution to this problem (Yuan et al., 2022; Yin et al., 2019).

Impressive earlier methods use Markov Random Fields (MRF) to model monocular cues and the relation between several over-segmented image parts (Saxena et al., 2007; 2008). Nevertheless, with the recent surge in neural network architectures (Krizhevsky et al., 2012; Simonyan & Zisserman, 2015; He et al., 2016), which has an extraordinary capability to perform complex regression, many current works use deep learning to solve SIDP and have demonstrated high-quality results (Yuan et al., 2022; Aich et al., 2021; Bhat et al., 2021; Eigen et al., 2014; Fu et al., 2018; Lee et al., 2019;

---

\*Corresponding Author
[1]
[2]For official code refer here

2021). Popular recent methods for SIDP are mostly supervised. But even then, they are used less in real-world applications than geometric multiple view methods (Labbé & Michaud, 2019; Müller et al., 2022). Nonetheless, a good solution to SIDP is highly desirable in robotics (Yang et al., 2020), virtual-reality (Hoiem et al., 2005), augmented reality (Du et al., 2020), view synthesis (Hoiem et al., 2005) and other related vision tasks (Liu et al., 2019).

In this paper, we advocate that despite the supervised approach being encouraging, SIDP advancement should not wholly rely on the increase of dataset sizes. Instead, geometric cues and scene priors could help improve the SIDP results. Not that scene priors have not been studied to improve SIDP accuracy in the past. For instance, Chen et al. (2016) uses pairwise ordinal relations between points to learn scene depth. Alternatively, Yin et al. (2019) uses surface normals as an auxiliary loss to improve performance. Other heuristic approaches, such as Qi et al. (2018), jointly exploit the depth-to-normal relation to recover scene depth and surface normals. Yet, such state-of-the-art SIDP methods have limitations: for example, the approach in Chen et al. (2016) - using ordinal relation to learn depth - over-smooths the depth prediction results, thereby failing to preserve high-frequency surface details. Conversely, Yin et al. (2019) relies on good depth map prediction from a deep network and the idea of virtual normal. The latter is computed by randomly sampling three non-collinear points with large distances. This is rather complex and heuristic in nature. Qi et al. (2018) uses depth and normal consistency, which is good, yet it requires good depth map initialization.

This brings us to the point that further generalization of the regression-based SIDP pipeline is required. As mentioned before, existing approaches in this direction have limitations and are complex. In this paper, we propose a simple approach that provides better depth accuracy and generalizes well across different scenes. To this end, we resort to the physics of variation (Mollenhoff et al., 2016; Chambolle et al., 2010) in the neural network design for better generalization of the SIDP network, which by the way, keeps the essence of affine invariance (Yin et al., 2019). An image of a general scene—indoor or outdoor, has a lot of spatial regularity. And therefore, introducing a variational constraint provides a convenient way to ensure spatial regularity and to preserve information related to the scene discontinuities (Chambolle et al., 2010). Consequently, the proposed network is trained in a fully-supervised manner while encouraging the network to be mindful of the scene regularity where the variation in the depth is large (cf. Sec.3.1). In simple terms, depth regression must be more than parameter fitting, and at some point, a mindful decision must be made—either by imaging features or by scene depth variation, or both. As we demonstrate later in the paper, such an idea boosts the network's depth accuracy while preserving the high-frequency and low-frequency scene information (see Fig.1).

Our neural network for SIDP disentangles the absolute scale from the metric depth map. It models an unscaled depth map as the optimal solution to the pixel-level variational constraints via weighted first-order differences, respecting the neighboring pixel depth gradients. Compared to previous methods, the network's task has been shifted away from pixel-wise metric depth learning to learning the first-order differences of the scene, which alleviates the scale ambiguity and favors scene regularity. To realize that, we initially employ a neural network to predict the first-order differences of the depth map. Then, we construct the partial differential equations representing the variational constraints by reorganizing the differences into a large matrix, i.e., an over-determined system of equations. Further, the network learns a weight matrix to eliminate redundant equations that do not favor the introduced first-order difference constraint. Finally, the closed-form depth map solution is recovered via simple matrix operations.

When tested on the KITTI (Geiger et al., 2012) and NYU Depth V2 (Silberman et al., 2012) test sets, our method outperforms prior art depth prediction accuracy by a large margin. Moreover, our model pre-trained on NYU Depth V2 better generalizes to the SUN RGB-D test set.

## 2 PRIOR WORK

Depth estimation is a longstanding task in computer vision. In this work, we focus on a fully-supervised, single-image approach, and therefore, we discuss prior art that directly relates to such approach. Broadly, we divide the popular supervised SIDP methods into three sub-categories.

***(i)* Depth Learning using Ranking or Ordinal Relation Constraint.** Zoran et al. (2015) and Chen et al. (2016) argue that the ordinal relation between points is easier to learn than the metric depth. To this end, Zoran et al. (2015) proposes constrained quadratic optimization while Chen

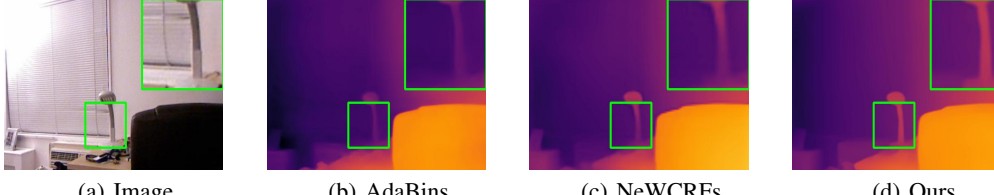

|            |              |               |           |
| :--------: | :----------: | :-----------: | :-------: |
| (a) Image  | (b) AdaBins  | (c) NeWCRFs   | (d) Ours  |

Figure 1: Qualitative comparison of our method's depth result with recent state-of-the-art methods such as AdaBins (Bhat et al., 2021), NeWCRFs (Yuan et al., 2022) on NYU Depth V2 test set (Silberman et al., 2012). It can be observed that our method predicts high-frequency details better than other recent methods.

et al. (2016) relies on the variation of the inception module to solve the problem. Later, Xian et al. (2020) proposes structure-guided sampling strategies for point pairs to improve training efficiency. Recently, Lienen et al. (2021) elaborates on the use of listwise ranking method based on the Plackett-Luce model (Luce, 2012). The drawback of such approaches is that the ordinal relationship and ranking over smooth the depth solution making accurate metric depth recovery challenging.

*(ii)* **Depth Learning using Surface Normal Constraint.** Hu et al. (2019) introduces normal loss in addition to the depth loss to overcome the distorted and blurry edges in the depth prediction. Yin et al. (2019) proposes the concept of virtual normal to impose 3D scene constraint explicitly and to capture the long-range relations in the depth prediction. The long-range dependency in 3D is introduced via random sampling of three non-colinear points at a large distance from the virtual plane. Lately, Long et al. (2021) proposes an adaptive strategy to compute the local patch surface normals at train time from a set of randomly sampled candidates and overlooks it during test time.

*(iii)* **Depth Learning using other Heuristic Refinement Constraint.** There has been numerous works attempt to refine the depth prediction as a post-processing step. Liu et al. (2015), Li et al. (2015) and Yuan et al. (2022) propose to utilize the Conditional Random Fields (CRF) to smooth the depth map. Lee et al. (2019) utilizes the planar assumptions to regularize the predicted depth map. Qi et al. (2018) adopts an auxiliary network to predict the surface normal, and then refine the predicted depth map following their proposed heuristic rules. There are mainly two problems with such approaches: Firstly, these approaches rely on a good depth map initialization. Secondly, the heuristic rules and the assumptions might result in over-smoothed depth values at objects boundaries.

Meanwhile, a few works, such as Ramamonjisoa et al. (2020), Cheng et al. (2018), Li et al. (2017) were proposed in the past with similar inspirations. Ramamonjisoa et al. (2020), Cheng et al. (2018) methods are generally motivated towards depth map refinement predicted from an off-the-shelf network. On the other hand, Cheng et al. (2018) proposes to use an affinity matrix that aims to learn the relation between each pixel's depth value and its neighbors' depth values. However, the affinity matrix has no explicit supervision, which could lead to imprecise learning of neighboring relations providing inferior results. On the contrary, our approach is mindful of imposing the first-order difference constraint leading to better performance. Earlier, Li et al. (2017) proposed two strategies for SIDP, i.e., fusion in an end-to-end network and fusion via optimization. The end-to-end strategy fuses the gradient and the depth map via convolution layers without any constraint on convolution weights, which may not be an apt choice for a depth regression problem such as SIDP. On the other hand, the fusion via optimization strategy is based on a non-differentiable strategy, leading to a non-end-to-end network loss function. Contrary to that, our method is well-constrained and performs quite well with a loss function that helps end-to-end learning of our proposed network. Not long ago, Lee & Kim (2019) proposed to estimate relative depths between pairs of images and ordinary depths at a different scale. By exploiting the rank-1 property of the pairwise comparison matrix, it recovers the relative depth map. Later, relative and ordinary depths are decomposed and fused to recover the depth. On a slightly different note, Lee & Kim (2020) studies the effectiveness of various losses and how to combine them for better monocular depth prediction.

To sum up, our approach allows learning of confidence weight to select reliable gradient estimation in a fully differentiable manner. Further, it proffers the benefits of the variational approach to overcome the limitations of the existing state-of-the-art methods. More importantly, the proposed method can provide excellent depth prediction without making extra assumptions such as good depth initialization, piece-wise planar scene, and assumptions used by previous works mentioned above.

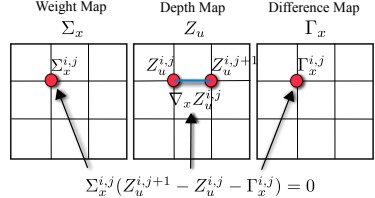 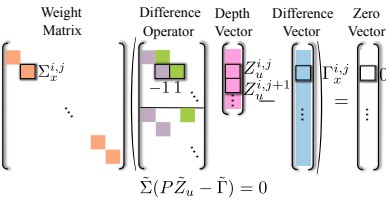

(a) First order depth variation along $x$-axis.    (b) Overall matrix by ordering the terms.

Figure 2: **Illustration of the idea**. (a) Depth gradient constraint along $x$ axis at location $(i, j)$ in $4 \times 4$ matrix form. (b) Construction of the overall matrix formulation with constraints at all the pixel locations.

## 3 METHOD

In this section, we first describe our proposed variational constraint and then present the overall network architecture leading to the overall loss function.

### 3.1 VARIATIONAL CONSTRAINT

Here we introduce our variational constraint and how it can be useful for depth estimation. Consider an unscaled depth map as $Z_u \in \mathbb{R}^{H \times W}$, with $(H, W)$ symbolizing the height and width, respectively. Assuming $\Gamma_x \in \mathbb{R}^{H \times W}$ and $\Gamma_y \in \mathbb{R}^{H \times W}$ as the gradient of $Z_u$ in the $x$ and $y$ axis, we write

$$\nabla Z_u = [\Gamma_x, \ \Gamma_y]^{\mathrm{T}}. \tag{1}$$

Here, $x$ and $y$ subscript corresponds to the direction from left to right ($x$-axis) and top to bottom of the image ($y$-axis), respectively. Elaborating on this, we can write

$$\Gamma_x^{i,j} = \nabla_x Z_u^{i,j} = Z_u^{i,j+1} - Z_u^{i,j}; \ \ \Gamma_y^{i,j} = \nabla_y Z_u^{i,j} = Z_u^{i+1,j} - Z_u^{i,j}. \tag{2}$$

Suppose we augment Eq.(2) expression for all $(i, j)$, $i \in \{1, ..., H\}$ and $j \in \{1, ..., W\}$. In that case, we will end up with an over-determined system with $2HW$ equations in total. Given the predicted $\Gamma_x$ and $\Gamma_y$, we aim to recover the $HW$ unknown variables in $Z_u$. However, some of the equations could be spurious and deteriorate the overall depth estimation result rather than improving it. As a result, we must be mindful about selecting the equation that respects the imposed first-order constraint and maintains the depth gradient to have a meaningful fitting for better generalization. To that end, we introduce confidence weight $\Sigma_x \in [0, 1]^{H \times W}, \Sigma_y \in [0, 1]^{H \times W}$ for gradient along $x, y$ direction. Consequently, we multiply the above two equations by the confidence weight term $\Sigma_x^{i,j}$ and $\Sigma_y^{i,j}$, respectively. On one hand, if the confidence is close to 1, the equation will have priority to be satisfied by the optimal $Z_u$. On the other hand, if the confidence is close to 0, we must ignore the equation. For better understanding, we illustrate the first-order difference and weighted matrix construction in Fig.2 (a) and Fig.2 (b).

Next, we reshape the $\Sigma_x, \Sigma_y, \Gamma_x, \Gamma_y$, and $Z_u$ into column vectors $\tilde{\Sigma}_x \in [0, 1]^{HW \times 1}, \tilde{\Sigma}_y \in [0, 1]^{HW \times 1}, \tilde{\Gamma}_x \in \mathbb{R}^{HW \times 1}, \tilde{\Gamma}_y \in \mathbb{R}^{HW \times 1}$, and $\tilde{Z}_u \in \mathbb{R}^{HW \times 1}$, respectively. Organizing $\tilde{\Sigma} = \mathbf{diag}([\tilde{\Sigma}_x; \tilde{\Sigma}_y]) \in \mathbb{R}^{2HW \times 2HW}$ and $\tilde{\Gamma} = \mathbf{concat}[\tilde{\Gamma}_x; \tilde{\Gamma}_y] \in \mathbb{R}^{2HW \times 1}$, we can write the overall expression in a compact matrix form using simple algebra as follows

$$\tilde{\Sigma} P \tilde{Z}_u = \tilde{\Sigma} \tilde{\Gamma} \tag{3}$$

where $P \in \{1, 0, -1\}^{2HW \times HW}$ is the first-order difference operator. Specifically, $P$ is a sparse matrix with only a few elements as 1 or -1. The $i^{\text{th}}$ row of $P$ provides the first-order difference operator for the $i^{\text{th}}$ equation. The position of 1 and -1 indicates which pair of neighbors to be considered for the constraint. Fig.2 (b) provides a visual intuition about this matrix equation.

Eq.(3) can be utilized to recover $Z_u$ from the predicted $\Gamma_x, \Gamma_y, \Sigma_x$, and $\Sigma_y$. As alluded to above, we have more equations than unknowns, hence, we resort to recovering the optimal depth map

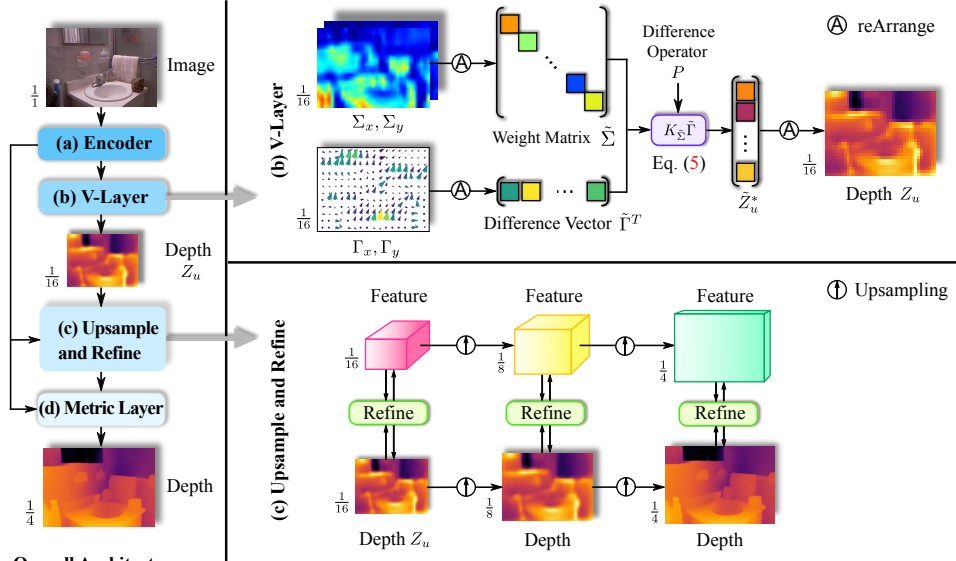

Figure 3: **Overview of our framework.** Given an input image, first an encoder is employed to extract features. Then we predict the depth map by the V-layer. Next, we gradually upsample and refine the depth map. In the end, we recover the metric depth by the metric layer.

$\tilde{Z}_u^* \in \mathbb{R}^{HW \times 1}$ by minimizing the following equation:

$$\tilde{Z}_u^* = \arg\min_{\tilde{Z}_u} ||\tilde{\Sigma}(P\tilde{Z}_u - \tilde{\Gamma})||_2. \tag{4}$$

Refer Appendix for the full derivation. The closed-form solution can be written as follows:

$$\tilde{Z}_u^* = \overbrace{(P^T\tilde{\Sigma}^2 P)^{-1}P^T\tilde{\Sigma}^2}^{K_{\tilde{\Sigma}}}\tilde{\Gamma}. \tag{5}$$

Denote $K_{\tilde{\Sigma}} \triangleq (P^T\tilde{\Sigma}^2 P)^{-1}P^T\tilde{\Sigma}^2$ in Eq.(5), we write overall equation as $\tilde{Z}_u^* = K_{\tilde{\Sigma}}\tilde{\Gamma}$. Next, we describe the overall network architecture.

## 3.2 Overall Network Architecture

Our overall network architecture is composed of four main modules as follows.

*(a)* **Encoder.** Given an input image, the encoder computes the hierarchical feature maps through a series of stages. To be precise, our encoder has four stages. Each stage contains transformer blocks (Liu et al., 2021b). At the end of each stage, we collect the final feature map as the output of the encoder resulting in the encoded feature maps with strides 4, 8, 16, and 32, respectively. Our encoder module is inspired by Liu et al. (2021b), a recent state-of-the-art transformer network design. We use it as our backbone by removing the final global pooling layer and fully connected layer.

*(b)* **Variational Layer (V-Layer).** The goal of this layer is to compute a map from encoded feature maps to unscaled depth map, which adheres to the first-order variational constraint. As of V-layer, we feed the feature maps of strides 16 and 32 as input which is the output of the encoder. Since these features are at different resolutions, we upsample the feature map of stride 32 to stride 16 via bi-linear interpolation and concatenate to construct $I_\Phi \in \mathbb{R}^{C \times H \times W}$, where $(H, W, C)$ symbolizing the height, width, and the number of channels, respectively. Note that $H, W$ is not the same as the original resolution of the ground-truth depth and images. We use of two convolutional layers on $I_\Phi$ to predict the depth gradient and corresponding weight for each pixel as follows:

$$\{\Gamma_x, \Gamma_y, \Sigma_x, \Sigma_y\} = f(I_\Phi; \theta) \tag{6}$$

where, $f(I_\Phi; \theta)$ denotes the convolutional layers with parameters $\theta$. The predicted depth gradients $\Gamma_x$ and $\Gamma_y$ are observed to be more accurate at smooth surface than at boundaries. This brings

us again to the point made above that we must take care of which first-order constraint must be included and discarded during regression. Using the Eq.(6) prediction, we construct the variational constraint Eq.(3), and obtain the unscaled depth map following Eq.(5). The resulting depth map has a resolution of 1/16 to the original image, which is later upsampled to the appropriate resolution.

To capture more scene features, we generate multiple channels (denoted as $S$) of $\{\Gamma_x, \Gamma_y, \Sigma_x, \Sigma_y\}$ using Eq.(6). As a result, we have a group of depth maps stacked along the channel dimension. For a feature map with spatial resolution $H \times W$, our V-layer has a complexity of $O(H^3 W^3)$. To overcome complexity issue, we perform V-layer operation on feature maps with stride 16 and then upsample and refine the depth maps in the later stage. The V-layer pipeline is shown in Fig.3(b).

*(c)* **Upsample and Refine.** This module upsamples and refines the input depth map via encoded features at a given depth map resolution. To this end, we perform refinement at three different resolutions in a hierarchical manner. Given the V-layer depth map at 1/16 resolution, we first refine the depth via encoded features at this resolution. Concretely, this refinement is done using the following set of operations. (1) concatenate the feature map and the depth map; (2) use one convolutional layer with ReLU activation to fuse the feature and depth information; and (3) predict refined feature and depth map via a convolutional layer. Later, the refined feature and depth map are upsampled and fed into 1/8 for later refinement using the same above set of operations. Finally, the exact is done at 1/4 resolution. Note that these steps are performed in a sequel. At the end of this module, we have a depth map of 1/4 of the actual resolution. The upsample and refine procedure are shown in Fig.3(c).

*(d)* **Metric Layer.** We must infer the global scene scale and shift to recover the metric depth. For this, we perform global max pooling on the encoded feature map of stride 32. The resulting vector is fed into a stack of fully connected layers to regress the two scalars, i.e., one representing the scale and while other representing the shift. Using the feature map of stride 32 is motivated by the observation that we have a much richer global scene context using it than at higher depth resolution. It also provides a good compromise between computational complexity and accuracy.

### 3.3 LOSS FUNCTION

**Depth Loss.** It estimates the scale-invariant difference between the ground-truth depth and prediction at train time (Eigen et al., 2014). The difference is computed by upsampling the predicted depth map to the same resolution as the ground truth via bi-linear interpolation. Denoting the predicted and ground-truth depth as $\hat{Z} \in \mathbb{R}^{m \times n}$, $Z_{gt} \in \mathbb{R}^{m \times n}$ we compute the depth loss as follows

$$\mathcal{L}_{depth}(\hat{Z}, Z_{gt}) = \frac{1}{N} \sum_{(i,j)} (e^{i,j})^2 - \frac{\alpha}{N^2} (\sum_{i,j} e^{i,j})^2, \text{ where}, e^{i,j} = \log \hat{Z}^{i,j} - \log Z_{gt}^{i,j}. \quad (7)$$

Here, $N$ is the number of positions with valid measurements and $\alpha \in [0, 1]$ is a hyper-parameter. Note that the above loss is used for valid measurements only.

**Variational Loss.** We define this loss using the output of V-layer. Suppose the ground-truth depth map to be $Z_{gt} \in \mathbb{R}^{m \times n}$ and the predicted depth map for $S$ channels as $Z_u \in \mathbb{R}^{S \times H \times W}$. Since the depth resolution is not same at this layer, we downsample the ground truth. It is observed via empirical study that low-resolution depth map in fact help capture the first-order variational loss among distant neighbors. Accordingly, we downsample the $Z_{gt}$ instead of upsamping $Z_u$. We downsample $Z_{gt}$ denoted as $Q_{gt} \in \mathbb{R}^{H \times W}$ by random pooling operation, i.e., we randomly select a location where we have a valid measurement since ground-truth data may have pixels with no depth values. The coordinates of selected location in $Z_{gt} \mapsto Z_u \in \mathbb{R}^{S \times H \times W}$ and the corresponding depth value is put in $\hat{Q} \in \mathbb{R}^{S \times H \times W}$ via bi-linear interpolation. We compute the variational loss as

$$\mathcal{L}_{var}(\hat{Q}, Q_{gt}) = \frac{1}{N'} \sum_{(i,j)} |\text{Conv}(\hat{Q})^{i,j} - \nabla Q_{gt}^{i,j}| \quad (8)$$

where $N'$ is the number of positions having valid measurements, $\nabla$ symbolises the first-order difference operator, and $\text{Conv}$ refers to the convolutional layer. Here, we use the $\text{Conv}$ layer to fuse $S$ depth maps into a single depth map and also to compute its horizontal and vertical gradient.

**Total Loss.** We define the total loss as the sum of the depth loss and the variational loss i.e., $\mathcal{L} = \mathcal{L}_{depth} + \lambda \mathcal{L}_{var}$, where $\lambda$ is the regularization parameter set to 0.1 for all our experiments.

Table 1: Comparison with the state-of-the-art methods on the NYU test set (Silberman et al., 2012). Please refer to Sec.4.1 for details.

| Method | Backbone | SILog $\downarrow$ | Abs Rel $\downarrow$ | RMS$\downarrow$ | RMS log$\downarrow$ | $\delta_1 \uparrow$ | $\delta_2 \uparrow$ |
|---|---|---|---|---|---|---|---|
| GeoNet (Qi et al., 2018) | ResNet-50 | - | 0.128 | 0.569 | - | 0.834 | 0.960 |
| DORN (Fu et al., 2018) | ResNet-101 | - | 0.115 | 0.509 | - | 0.828 | 0.965 |
| VNL (Yin et al., 2019) | ResNeXt-101 | - | 0.108 | 0.416 | - | 0.875 | 0.976 |
| TransDepth (Yang et al., 2021) | ViT-B | - | 0.106 | 0.365 | - | 0.900 | 0.983 |
| ASN (Long et al., 2021) | HRNet-48 | - | 0.101 | 0.377 | - | 0.890 | 0.982 |
| BTS (Lee et al., 2019) | DenseNet-161 | 11.533 | 0.110 | 0.392 | 0.142 | 0.885 | 0.978 |
| DPT-Hybrid (Ranftl et al., 2021) | ViT-B | - | 0.110 | 0.357 | - | 0.904 | 0.988 |
| AdaBins (Bhat et al., 2021) | EffNet-B5+ViT-mini | 10.570 | 0.103 | 0.364 | 0.131 | 0.903 | 0.983 |
| ASTrans (Chang et al., 2021) | ViT-B | 10.429 | 0.103 | 0.374 | 0.132 | 0.902 | 0.985 |
| NeWCRFs (Yuan et al., 2022) | Swin-L | 9.102 | 0.095 | 0.331 | 0.119 | 0.922 | **0.992** |
| **Ours** | Swin-L | **8.198** | **0.086** | **0.304** | **0.108** | **0.937** | **0.992** |
| **% Improvement** | | -9.93% | -9.47% | -8.16% | -9.24% | +1.63% | +0.00% |

Table 2: Comparison with the state-of-the-art methods on the the KITTI official test set (Geiger et al., 2012). We only list the results from the published methods. Please refer to Sec.4.1 for details.

| Method | Backbone | SILog$\downarrow$ | Abs Rel$\downarrow$ | Sq Rel$\downarrow$ | iRMS$\downarrow$ |
|---|---|---|---|---|---|
| DLE (Liu et al., 2021a) | ResNet-34 | 11.81 | 9.09 | 2.22 | 12.49 |
| DORN (Fu et al., 2018) | ResNet-101 | 11.80 | 8.93 | 2.19 | 13.22 |
| BTS (Lee et al., 2019) | DenseNet-161 | 11.67 | 9.04 | 2.21 | 12.23 |
| BANet (Aich et al., 2021) | DenseNet-161 | 11.55 | 9.34 | 2.31 | 12.17 |
| PWA (Lee et al., 2021) | ResNeXt-101 | 11.45 | 9.05 | 2.30 | 12.32 |
| ViP-DeepLab (Qiao et al., 2021) | - | 10.80 | 8.94 | 2.19 | 11.77 |
| NeWCRFs (Yuan et al., 2022) | Swin-L | 10.39 | 8.37 | 1.83 | 11.03 |
| **Ours** | Swin-L | **9.84** | **7.96** | **1.66** | **10.44** |
| **% Improvement** | | -5.29% | -4.90% | -9.29% | -5.35% |

## 4 EXPERIMENTS AND RESULTS

**Implementation Details** We implemented our method in PyTorch 1.7.1 (Python 3.8) with CUDA 11.0. The software is evaluated on a computing machine with Quadro-RTX-6000 GPU.

**Datasets.** We performed experiments on three benchmark datasets namely NYU Depth V2 (Silberman et al., 2012), KITTI (Geiger et al., 2012), and SUN RGB-D (Song et al., 2015). *(a)* **NYU Depth V2** contains images with $480 \times 640$ resolution with depth values ranging from 0 to 10 meters. We follow the train and test set split from Lee et al. (2019), which contains 24,231 train images and 654 test images. *(b)* **KITTI** contains images with $352 \times 1216$ resolution where depth values range from 0 to 80 meters. The official split provides 42,949 train, 1,000 validation, and 500 test images. Eigen et al. (2014) provides another train and test set split for this dataset which has 23,488 train and 697 test images. *(c)* **SUN RGB-D** We preprocess its images to $480 \times 640$ resolution for consistency. The depth values range from 0 to 10 meters. We use the official test set (5050 images) for evaluation.

**Training Details.** We use (Liu et al., 2021b) network as our backbone, which is pre-trained on ImageNet (Deng et al., 2009). We use the Adam optimizer (Kingma & Ba, 2014) without weight decay. We decrease the learning rate from $3e^{-5}$ to $1e^{-5}$ by the cosine annealing scheduler. To avoid over-fitting, we augment the images by horizontal flipping. For KITTI (Geiger et al., 2012), the model is trained for 10 epochs for the official split and 20 epochs for the Eigen split (Eigen et al., 2014). For NYU Depth V2 (Silberman et al., 2012), the model is trained for 20 epochs.

**Evaluation Metrics.** We report statistical results on popular evaluation metrics such as square root of the Scale Invariant Logarithmic error (**SILog**), Relative Squared error (**Sq Rel**), Relative Absolute Error (**Abs Rel**), Root Mean Squared error (**RMS**), and threshold accuracy. Mathematical definition related to each one of them is provided in the Appendix.

### 4.1 COMPARISON TO STATE OF THE ART

Tab.(1), Tab.(2), Tab.(3), and Tab.(4) provide statistical comparison results with the competing methods on NYU Depth V2, KITTI official split, KITTI Eigen split, and SUN RGB-D, respectively. Our proposed approach shows the best results for all the evaluation metrics. Particularly on the NYU test set, we reduce the SILog error from the previous best result, 9.102 to 8.198, and increase $\delta_1$ from 0.922 to 0.937. More qualitative results including V-layer output are presented in the Appendix.

Table 3: Comparison with the state-of-the-art methods on the KITTI Eigen test set (Eigen et al., 2014).

| Method | Backbone | SILog ↓ | Abs Rel ↓ | RMS↓ | RMS log↓ | $\delta_1$ ↑ | $\delta_2$ ↑ |
|---|---|---|---|---|---|---|---|
| DORN (Fu et al., 2018) | ResNet-101 | - | 0.072 | 0.273 | 0.120 | 0.932 | 0.984 |
| VNL (Yin et al., 2019) | ResNeXt-101 | - | 0.072 | 0.326 | 0.117 | 0.938 | 0.990 |
| TransDepth (Yang et al., 2021) | ViT-B | 8.930 | 0.064 | 0.275 | 0.098 | 0.956 | 0.994 |
| BTS (Lee et al., 2019) | DenseNet-161 | 8.933 | 0.060 | 0.280 | 0.096 | 0.955 | 0.993 |
| DPT-Hybird (Ranftl et al., 2021) | ViT-B | - | 0.062 | 0.257 | - | 0.959 | 0.995 |
| AdaBins (Bhat et al., 2021) | EffNet-B5+ViT-mini | 8.022 | 0.058 | 0.236 | 0.089 | 0.964 | 0.995 |
| ASTrans (Chang et al., 2021) | ViT-B | 7.897 | 0.058 | 0.269 | 0.089 | 0.963 | 0.995 |
| NeWCRFs (Yuan et al., 2022) | Swin-L | 6.986 | 0.052 | 0.213 | 0.079 | 0.974 | **0.997** |
| **Ours** | Swin-L | **6.817** | **0.050** | **0.209** | **0.076** | **0.977** | **0.997** |
| **% Improvement** | | -2.42% | -3.85% | -1.88% | -3.80% | +0.03% | +0.00% |

Table 4: Comparison with AdaBins and NeWCRFs on SUN RGB-D test set. All methods are trained on NYU Depth V2 train set without fine-tuning on SUN RGB-D.

| Method | Backbone | SILog ↓ | Abs Rel ↓ | RMS↓ | RMS log↓ | $\delta_1$ ↑ | $\delta_2$ ↑ |
|---|---|---|---|---|---|---|---|
| AdaBins(Bhat et al., 2021) | EffNet-B5+ViT-mini | 13.652 | 0.110 | 0.321 | 0.137 | 0.906 | 0.982 |
| NeWCRFs (Yuan et al., 2022) | Swin-L | 13.695 | 0.105 | 0.322 | 0.138 | 0.920 | 0.980 |
| **Ours** | Swin-L | **12.596** | **0.094** | **0.299** | **0.127** | **0.929** | **0.983** |
| **% Improvement** | | -7.73% | -10.48% | -6.85% | -7.30% | +0.98% | +0.10% |

For the SUN RGB-D test set, all competing models, including ours, are trained on the NYU Depth V2 training set (Silberman et al., 2012) *without* fine-tuning on the SUN RGB-D. In addition, we align the predictions from all the models with the ground truth by a scale and shift following Ranftl et al. (2020). Tab.(4) results show our method's better generalization capability than other approaches. Extensive visual results are provided in the Appendix and supplementary video.

## 4.2 ABLATION STUDY

All the ablation presented below is conducted on NYU Depth V2 test set (Silberman et al., 2012).

*(i)* **Effect of V-Layer.** To understand the benefit and outcome of our variational layer compared to other popular alternative layers in deep neural networks for this problem, we performed this ablation study. We replace our V-layer firstly with a convolutional layer and later with a self-attention layer. Tab.(5) provides the depth prediction accuracy for this ablation. For each introduced layer in Tab.(5), the first and second rows show the performance of the depth map predicted *with* (w) and *without* (w/o) subsequent refinements (cf. Sec.3.2 (c)), respectively. For the self-attention layer, we follow the ViT (Dosovitskiy et al., 2021) and set the patch size to be one as we use the feature map with stride 16. We also adopt the learnable position embedding (PE) with 128 dimensions. We set the number of heads to be 4 and the number of hidden units to be 512. As shown in Tab.(5), our V-layer indeed helps improve the accuracy of depth prediction compared to other well-known layers. More experiments on KITTI and SUN RBG-D are provided in the Appendix.

*(ii)* **Performance with Different Network Backbone.** We evaluate the effects of our V-layer with different types of network backbones. For this ablation, we use Swin-Large (Liu et al., 2021b), Swin-Small (Liu et al., 2021b), and ConvNeXt-Small (Liu et al., 2022). The SILog error is shown in Fig. 4. The results show that our V-layer improves the transformer and the convolutional network performance. An important observation is that our V-layer shows excellent improvements in depth prediction accuracy on weaker network backbones.

*(iii)* **Performance with Change in the Value of** $S$**.** For this ablation, we change the value of $S$ in the V-layer and observe its effects (cf. Sec.3.2 (b)). By increasing $S$, we generate more channels of $\tilde{\Gamma}$ and $\tilde{\Sigma}$ which in-effect increases V-layer parameters. In the subsequent step, we expand the

Table 5: **Benefit of V-layer.** We replace the proposed V-layer with a single convolutional layer and a self-attention layer, and evaluate the accuracy of depth map predicted with and without subsequent refinements.

| Layer | Refine | SILog ↓ | Abs Rel ↓ | RMS↓ | RMS log ↓ | $\delta_1$ ↑ | $\delta_2$ ↑ |
|---|---|---|---|---|---|---|---|
| Convolution | w/o | 8.830 | 0.090 | 0.325 | 0.114 | 0.927 | 0.990 |
| | w/ | 8.688 | 0.089 | 0.317 | 0.113 | 0.928 | 0.991 |
| Self-Attention + PE | w/o | 8.790 | 0.090 | 0.318 | 0.114 | 0.927 | 0.990 |
| | w/ | 8.595 | 0.089 | 0.316 | 0.112 | 0.929 | 0.991 |
| **V-Layer** | w/o | 8.422 | 0.087 | 0.308 | 0.110 | 0.936 | 0.990 |
| | **w/** | **8.198** | **0.086** | **0.304** | **0.108** | **0.937** | **0.992** |

number of channels to 128 by a convolutional layer to use the subsequent layers as they are. The results are shown in Tab.(6). For reference, we also present the result by replacing the V-layer with a convolutional layer in the first row in Tab.(6). By increasing $S$, we reduce the SILog error, at the price of the speed (FPS). Yet, no real benefit is observed with $S$ more than 16.

*(iv)* **Effect of Confidence Weight Matrix & Difference Operator in V-Layer.** For this ablation, we study the network's depth prediction under four different settings. (a) without V-layer and replace it with convolutional layer (b) without the confidence weight matrix (c) with learnable difference operator and (d) our full model. The depth prediction accuracy observed under these settings is provided in Tab.(7). Clearly, our full model has better accuracy. An important empirical observation, we made during this test is when we keep $P$ learnable V-layer has more learnable parameters, the performance becomes worse than with fixed difference operator.

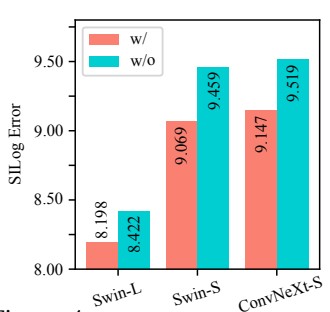

**Figure 4:** Evaluation on Swin-L, Swin-S, ConvNeXt-S **w/** and **w/o** the V-layer.

**Table 6:** Analysis of the number of feature groups. More groups reduce the SILog error.

|            | SILog↓ | Abs Rel↓ | RMS↓   | FPS ↑   |
|------------|--------|----------|--------|---------|
| w/o V-layer | 8.688  | 0.089    | 0.317  | **9.343** |
| 1          | 8.456  | 0.088    | 0.310  | 8.175   |
| **16**     | 8.198  | 0.086    | **0.304** | 7.032 |
| 128        | **8.172** | **0.085** | 0.309 | 3.320  |

**Table 7:** Analysis of the confidence weight matrix $\tilde{\Sigma}$ and the difference operator $P$.

|                      | SILog↓ | Abs Rel↓ | RMS ↓  |
|----------------------|--------|----------|--------|
| (a) w/o V-layer      | 8.688  | 0.089    | 0.317  |
| (b) w/o $\tilde{\Sigma}$ | 8.537 | 0.089    | 0.316  |
| (c) learnable $P$    | 8.355  | 0.088    | 0.310  |
| (d) **full**         | **8.198** | **0.086** | **0.304** |

## 4.3 NETWORK PROCESSING TIME & PARAMETERS

We compared our method's inference time and the number of model parameters to the AdaBins (Bhat et al., 2021) and the NeWCRFs (Yuan et al., 2022). The inference time is measured on the NYU Depth V2 test set with batch size 1. We have removed the ensemble tricks in AdaBins and NeWCRFs for an unbiased evaluation, resulting in a slight increase in SILog error as compared to Tab.(1) statistics. As is shown in Tab.(8), our method is faster and better than AdaBins and NeWCRFs using Swin-Small backbone. With the same backbone as the NeWCRFs, i.e., Swin-Large, we achieve much better depth prediction results. Hence, our method with Swin-Small backbone provides a better balance between accuracy, speed and memory foot-print.

**Table 8:** Comparison of the inference time and parameters to AdaBins and NeWCRFs on NYU Depth V2. We show our results using the Swin-Small and Swin-Large backbone.

|                | AdaBins (Bhat et al., 2021) | NeWCRFs (Yuan et al., 2022) | Ours (Small) | Ours (Large) |
|----------------|-----------------------------|-----------------------------|--------------|--------------|
| SILog Error ↓  | 10.651                      | 9.171                       | **9.069**    | **8.198**    |
| Speed (FPS) ↑  | 5.638                       | 10.551                      | **11.891**   | 7.032        |
| Param (M) ↓    | **75**                      | 258                         | **76**       | 249          |

## 5 CONCLUSION

In conclusion, a simple and effective approach for inferring scene depth from a single image is introduced. The proposed SIDP approach is shown to better exploit the rigid scene prior, which is generally overlooked by the existing neural network-based methods. Our approach does not make explicit assumptions about the scene other than the scene gradient regularity, which holds for typical indoor or outdoor scenes. When tested on popular benchmark datasets, our method shows significantly better results than the prior art, both qualitatively and quantitatively.

## 6 ACKNOWLEDGMENT

This work was partly supported by ETH General Fund (OK), Chinese Scholarship Council (CSC), and The Alexander von Humboldt Foundation.

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

# A  APPENDIX

## A.1  TRAINING DETAILS

We implement our framework in PyTorch (Paszke et al., 2019). We adopt the Swin-Large (Liu et al., 2021b) as our backbone to conduct ablation experiments and compare with the state of the arts. And the backbone is pre-trained on ImageNet-22K (Deng et al., 2009) by image classification. For training, we use the Adam optimizer (Kingma & Ba, 2014) without weight decay. We decrease the learning rate from $3e^{-5}$ to $1e^{-5}$ by the cosine annealing scheduler. To avoid over-fitting, we augment the images by horizontal flipping. For both train and test, we keep the resolution of images to be $352 \times 1216$ in KITTI (Geiger et al., 2012), and $480 \times 640$ in both NYU Depth V2 (Silberman et al., 2012) and SUN RGB-D (Song et al., 2015). In KITTI, the model is trained for 10 epochs for the official split and 20 epochs for the Eigen split (Eigen et al., 2014). In NYU Depth V2, the model is trained for 20 epochs. We set the batch size to be 4 and 8 respectively for ablation experiments and comparison to the state of the arts.

## A.2 EVALUATION METRICS

Suppose the predicted and ground-truth depth to be $\hat{Z} \in \mathbb{R}^{m \times n}$ and $Z_{gt} \in \mathbb{R}^{m \times n}$, respectively, and the number of valid pixels to be $N$. We follow the existing methods (Yuan et al., 2022) and utilize the following measures for quantitative evaluation:

- square root of the Scale Invariant Logarithmic error (**SILog**): $\frac{1}{N} \sum_{i,j} (e^{i,j})^2 - \frac{1}{N^2} (\sum_{i,j} e^{i,j})^2$, where $e^{i,j} = \log \hat{Z}^{i,j} - \log Z_{gt}^{i,j}$;

- Relative Squared error (**Sq Rel**): $\frac{1}{N} \sum_{i,j} (\hat{Z}^{i,j} - Z_{gt}^{i,j})^2 / Z_{gt}^{i,j}$;

- Relative Absolute Error (**Abs Rel**): $\frac{1}{N} \sum_{i,j} |\hat{Z}^{i,j} - Z_{gt}^{i,j}| / Z_{gt}^{i,j}$;

- Root Mean Squared error (**RMS**): $\frac{1}{N} \sqrt{\sum_{i,j} (\hat{Z}^{i,j} - Z_{gt}^{i,j})^2}$;

- Root Mean Squared Logarithmic error (**RMS log**): $\frac{1}{N} \sqrt{\sum_{i,j} (e^{i,j})^2}$;

- threshold accuracy ($\boldsymbol{\delta_k}$): percentage of $\hat{Z}^{i,j}$ s.t. $\max(\frac{\hat{Z}^{i,j}}{Z_{gt}^{i,j}}, \frac{Z_{gt}^{i,j}}{\hat{Z}^{i,j}}) < 1.25^k$.

## A.3 DERIVATION OF THE VARIATIONAL CONSTRAINT

In this part, we introduce the derivation of the variational constraint in more detail. Firstly, we present the detailed form of the difference operator $P$, the confidence weight matrix $\tilde{\Sigma}$, the difference vector $\tilde{\Gamma}$ and the depth vector $\tilde{Z}_u$, when the depth map $Z_u$ is at resolution $2 \times 2$. Secondly we show the derivation of the optimal solution $\tilde{Z}_u^*$.

**Overall Matrix Form.** Suppose the unscaled depth map to be $Z_u \in \mathbb{R}^{2 \times 2}$, the difference map to be $\Gamma_x \in \mathbb{R}^{2 \times 2}$ and $\Gamma_y \in \mathbb{R}^{2 \times 2}$, and the corresponding confidence weight map to be $\Sigma_x \in [0, 1]^{2 \times 2}$ and $\Sigma_y \in [0, 1]^{2 \times 2}$. We first reorganize the elements in $\Gamma_x, \Gamma_y, \Sigma_x, \Sigma_y$ to construct the difference vector $\tilde{\Gamma} \in \mathbb{R}^{8 \times 1}$ and the confidence weight matrix $\tilde{\Sigma} \in \mathbb{R}^{8 \times 8}$:

$$\tilde{\Gamma} = \begin{pmatrix} \Gamma_x^{1,1} \\ \Gamma_x^{1,2} \\ \Gamma_x^{2,1} \\ \Gamma_x^{2,2} \\ \Gamma_y^{1,1} \\ \Gamma_y^{1,2} \\ \Gamma_y^{2,1} \\ \Gamma_y^{2,2} \end{pmatrix}, \tilde{\Sigma} = \begin{pmatrix} \Sigma_x^{1,1} & 0 & 0 & 0 & 0 & 0 & 0 & 0 \\ 0 & \Sigma_x^{1,2} & 0 & 0 & 0 & 0 & 0 & 0 \\ 0 & 0 & \Sigma_x^{2,1} & 0 & 0 & 0 & 0 & 0 \\ 0 & 0 & 0 & \Sigma_x^{2,2} & 0 & 0 & 0 & 0 \\ 0 & 0 & 0 & 0 & \Sigma_y^{1,1} & 0 & 0 & 0 \\ 0 & 0 & 0 & 0 & 0 & \Sigma_y^{1,2} & 0 & 0 \\ 0 & 0 & 0 & 0 & 0 & 0 & \Sigma_y^{2,1} & 0 \\ 0 & 0 & 0 & 0 & 0 & 0 & 0 & \Sigma_y^{2,2} \end{pmatrix}.$$

Next, we apply the difference operator $P \in \{1, 0, -1\}^{8 \times 4}$:

$$P = \begin{pmatrix} -1 & 1 & 0 & 0 \\ 0 & 0 & 0 & 0 \\ 0 & 0 & -1 & 1 \\ 0 & 0 & 0 & 1 \\ -1 & 0 & 1 & 0 \\ 0 & -1 & 0 & 1 \\ 0 & 0 & 0 & 0 \\ 0 & 0 & 0 & 1 \end{pmatrix}$$

on the depth vector $\tilde{Z}_u \in \mathbb{R}^{4 \times 1}$:

$$\tilde{Z}_u = \begin{pmatrix} Z_u^{1,1} \\ Z_u^{1,2} \\ Z_u^{2,1} \\ Z_u^{2,2} \end{pmatrix}.$$

Finally, we construct the following constraint equation:

$$\tilde{\Sigma} P \tilde{Z}_u = \tilde{\Sigma} \tilde{\Gamma}. \tag{9}$$

In the above equation, each row represents a weighted constraint for the first-order difference along the $x$-axis:

$$\Sigma_x^{i,j}(Z_u^{i,j+1} - Z_u^{i,j}) = \Sigma_x^{i,j}\Gamma_x^{i,j} \tag{10}$$

or along the $y$-axis:

$$\Sigma_y^{i,j}(Z_u^{i+1,j} - Z_u^{i,j}) = \Sigma_y^{i,j}\Gamma_y^{i,j} \tag{11}$$

where $(Z_u^{i,j+1} - Z_u^{i,j}) = \Gamma_x^{i,j}$ and $(Z_u^{i+1,j} - Z_u^{i,j}) = \Gamma_y^{i,j}$ are the variational constraints for the first-order difference, while $\Sigma_x^{i,j}$ and $\Sigma_y^{i,j}$ are the confidence weights for the constraints.

**Optimal Solution.** Given $\tilde{\Sigma}$ and $\tilde{\Gamma}$, we search for the optimal depth vector $\tilde{Z}_u^*$ by minimizing the residual of Eq.(9):

$$\tilde{Z}_u^* = \arg\min_{\tilde{Z}_u} ||\tilde{\Sigma}(P\tilde{Z}_u - \tilde{\Gamma})||_2. \tag{12}$$

Firstly, the objective function, $||\tilde{\Sigma}(P\tilde{Z}_u - \tilde{\Gamma})||_2$, is convex with respect to $\tilde{Z}_u$. Because $||\cdot||_2$ is a convex function. A composition of $||\cdot||_2$ and the affine function , $\tilde{\Sigma}(P\tilde{Z}_u - \tilde{\Gamma})$, is still convex.

Secondly, the optimal solution of a convex function can be found at where the first derivative is zero. Thereby we obtain the final solution $\tilde{Z}_u^* = (P^T\tilde{\Sigma}^2 P)^{-1}P^T\tilde{\Sigma}^2\tilde{\Gamma}$.

### A.4 NETWORK STRUCTURE DETAILS

In this part, we introduce the detailed structure of our network. In general we set the kernel size of convolutional layers to be 3 unless otherwise stated.

**(a) Encoder.** We adopt the Swin-Large (Liu et al., 2021b) as our backbone. The network first divides the image into patches each of size $4 \times 4$, and embeds each patch into a 96-dimensional vector. The above procedure is implemented by a convolutional layer with kernel size 4 and stride 4. Then there are 4 transformation stages to be applied to the embedded vector. The first, second, third and forth stages include $2, 2, 18, 2$ blocks, respectively. More specifically, each block will divide the feature map into non-overlapped windows of size $12 \times 12$ , and compute new features within each window following the transformer (Vaswani et al., 2017). The feature channels in the 4 stages are 192, 384, 768 and 1536, respectively, and the number of heads are 6, 12, 24, and 48, respectively. In the end of each stage, there will be a downsampling operation to reduce the resolution of the feature map by 2. We collect the feature map before the downsampling operation as the output of the stage. In the end, the strides of output feature maps are 4, 8, 16, and 32 respectively. And the channels are 192, 384, 768, and 1536, respectively.

**(b) Upsampling 32->16.** Given the feature maps of strides 32 and 16 from **(a)**, we first upsample the feature map of stride 32 to stride 16 via bi-linear interpolation. Then, we concatenate the feature maps and obtain a new feature map of stride 16 and channels $1536 + 768 = 2304$. We apply a convolutional layer that has 2304 input channels, 2304 output channels, and 4 groups, to fuse the information. We also append an instance normalization layer (Ulyanov et al., 2016) and a LeakyReLU activation function. Next, we apply another convolutional layer with 2304 input channels, 512 output channels to compress the feature channels. Again we append an instance normalization layer and the LeakyReLU activation function. In the end, we add a skip connection between the result feature map and the previous concatenated feature map by addition operation. The concatenated feature map will be transformed to consistent number of channels in advance by a convolutional layer with 2304 input channels and 512 output channels. The final feature map has stride 16 and 512 channels.

**(c) V-Layer.** The V-layer takes the output feature map from **(b)** as input. We first utilize a convolutional layer with 512 input channels and 512 output channels to transform the feature map into a more appropriate hidden space. The feature is transformed by the LeakyReLU activation function. Then we utilize another convolutional layer with 512 input channels and $2 \times 16 = 32$ output channels to predict the gradients along the horizontal and vertical axis, respectively. The number 16 represents that we predict 16 channels of horizontal gradients and 16 channels of vertical gradients respectively, where each channel is expected to capture different information in scenes. Similarly, we also use another convolutional layer with 512 input channels, $2 \times 16 = 32$ output channels to predict the corresponding confidence weight maps. The confidence weight will be transformed by the Sigmoid function. Then we reshape the predictions and compute the unscaled depth map for

each pair of gradient and confidence weight following Eq.(5). The unscaled depth maps computed from all the pairs will be concatenated along the channel. Thereby we obtain a depth map with 16 channels and stride 16. We apply a group normalization on the depth map, where the 16 channels are viewed as a single group. In the end, we expand the channels of the depth map into 128 by a convolutional layer. Thereby the output of the V-layer is a depth map with 128 channels and stride 16.

**(d) Refine 16.** We refine the feature map from **(b)** and the depth map from **(c)**. Specifically, we first concatenate the feature map with depth map, and obtain a new feature map with $512 + 128 = 640$ channels. Then we employ a convolutional layer with 640 input channels and 640 output channels to fuse the information. The result feature map will be further transformed by the LeakyReLU activation function. Next a convolutional layer with 640 input channels and 128 output channels is applied to predict the refined depth map. Similarly, we also predict the refined feature map by a convolutional layer with 640 input channels and 512 output channels.

**(e) Upsampling 16->8.** Here, we upsample the feature map from **(d)** to stride 8 via bi-linear interpolation. To recover the information of scenes, we concatenate with the feature map from the encoder that has the same stride. In such a way, we obtain a new feature map with $384 + 512 = 896$ channels and stride 8. Similar to **(b)**, we apply a convolutional layer with 896 input channels, 896 output channels, and 4 groups to fuse the information. Then the feature map is transformed by the instance normalization layer and the LeakyReLU activation function. Next, we apply another convolutional layer with 896 input channels and 256 output channels to compress feature channels. Again, we apply the instance normalization layer and LeakyReLU activation function. Same as **(b)**, we add a skip connection between the result feature map and the previous concatenated feature map by addition operation. The concatenated feature map will be transformed to the consistent number of channels by a convolutional layer with 896 input channels and 256 output channels. Thereby the final feature map has stride 8 and 256 channels.

**(f) Refine 8.** We refine the depth map from **(d)** and the feature map from **(e)**. The procedure is the same as **(d)**, except for the number of channels in convolutional layers are adapted accordingly. Thereby we introduce the pipeline briefly. We first upsample the depth map from **(d)** into stride 8 to concatenate with the feature map. Thereby we obtain a new feature map with $256 + 128 = 384$ channels. Then, we apply a convolutional layer with 384 input channels and 384 output channels to fuse the information. The feature map is then transformed by LeakyReLU activation function. Next, we utilize a convolutional layer with 384 input channels and 128 output channels to predict the refined depth map. Similarly, another convolutional layer with 384 input channels and 256 output channels is applied to predict the refined feature map.

**(g) Upsampling 8->4.** We upsample the feature map from **(f)** to stride 4. The procedure is the same as **(e)** except for the number of channels. More specifically, we first upsample the feature map and concatenate with the feature map from the encoder with consistent stride, obtaining a feature map with $256 + 192 = 448$ channels and stride 4. Then a convolutional layer with 448 input channels, 448 output channels, and 4 groups is applied to fuse the information. Then the result feature map is transformed by the instance normalization and LeakyReLU activation function. Next, we apply another convolutional layer with 448 input channels and 64 output channels to compress the feature channels. Again, the instance normalization layer and LeakyReLU function is applied. In the end, we add a skip connection between the final feature map and the previous concatenated feature map.

**(h) Refine 4.** We refine the depth map from **(f)** and the feature map from **(g)** following the same pipeline as **(d)**. We first upsample the depth map to stride 4, and concatenate with the feature map, obtaining a new feature map with $64 + 128 = 192$ channels and stride 4. Next, we apply a convolutional layer with 192 input channels and 192 output channels to fuse the information, and a LeakyReLU function to transform the feature map. In the end, we utilize another convolutional layer with 192 input channels and 128 output channels to predict the new depth map. The output is a depth map with 128 channels and stride 4.

**(i) Metric Layer.** We take the feature map with stride 32 from the encoder as input. We first apply global max pooling to compress the feature map into a vector with 1536 channels. Then we apply a fully-connected layer with 1536 input units and 384 output units to compress the channels. Next we apply a LeakyReLU function. In the end, we utilize another fully-connected layer with 384 input units and 2 output units to regress the scale and shift.

Table 9: Comparison with Ramamonjisoa et al. (2020), Cheng et al. (2018) and Li et al. (2017) on NYU Depth V2 test set (Silberman et al., 2012).

| Method | SILog ↓ | Abs Rel ↓ | RMS↓ | RMS log↓ | $\delta_1$ ↑ | $\delta_2$ ↑ |
|---|---|---|---|---|---|---|
| Ramamonjisoa et al. (2020) | 8.655 | 0.088 | 0.320 | 0.113 | 0.929 | 0.991 |
| Cheng et al. (2018) | 8.640 | 0.089 | 0.317 | 0.112 | 0.930 | 0.991 |
| Li et al. (2017) (end-to-end) | 8.557 | 0.088 | 0.315 | 0.112 | 0.931 | 0.991 |
| Li et al. (2017) (optimization) | 8.574 | 0.089 | 0.316 | 0.112 | 0.930 | 0.991 |
| **Ours** | **8.198** | **0.086** | **0.304** | **0.108** | **0.937** | **0.992** |

Table 10: **Impact of resolution**. We evaluate the performance of V-layer when operating on feature maps of stride 16 and 8, respectively.

| Stride | FPS↑ | SILog ↓ | Abs Rel ↓ | RMS↓ | RMS log↓ | $\delta_1$ ↑ | $\delta_2$ ↑ |
|---|---|---|---|---|---|---|---|
| 16 | **7.032** | 8.198 | **0.086** | **0.304** | **0.108** | **0.937** | **0.992** |
| 8 | 2.597 | **8.149** | **0.086** | 0.306 | **0.108** | 0.936 | **0.992** |

**(j) Final Prediction.** We take the output from **(d)**,**(f)**, **(h)** and **(i)** as input. We upsample all the depth maps to stride 1 via bi-linear interpolation, and fuse the depth maps into a single depth map by the addition operation and a convolutional layer with 128 input channels and 1 output channel. In the end, we add the depth map with the shift and multiply with the scale from **(i)**, respectively.

## A.5 STATISTICAL EVALUATION WITH OTHER METHODS

In this section, we compare with Ramamonjisoa et al. (2020), Cheng et al. (2018), and Li et al. (2017) on NYU Depth V2 (Silberman et al., 2012). More specifically, we apply their methods to refine the final predicted depth map. The displacement field, the affinity matrix, and the depth gradient are predicted from the refined feature map, which is the output of the last refinement module. For Cheng et al. (2018) we adopt their default hyper-parameters. The kernel size is 3, and the number of iterations is 24. Li et al. (2017) proposes the end-to-end fusion strategy and the optimization-based fusion strategy. For the end-to-end fusion strategy, we employ three convolutional layers with kernel size 5, and 16 channels. For the optimization-based fusion strategy, we use Adam optimizer. The learning rate is 0.01 and the number of iterations is 100. The results are shown in Tab. (9). Our method achieves better performance.

## A.6 ABLATION ON RESOLUTION

In this part, we evaluate the performance of the V-layer when applying on the feature map with stride 8. We achieve this by dividing the feature map into 4 rectangle sub-regions with equal area (top left, top right, bottom left and bottom right). We apply a V-layer on each sub-region. The predicted depth maps from the sub-regions will be stacked to recover the original resolution. The results are shown in Tab.(10). From stride 16 to 8, the improvement of accuracy is marginal. However, the inference speed (FPS) is significantly slow down.

## A.7 ABLATION ON KITTI AND SUN RGB-D

To further demonstrate the benefits of our V-layer on KITTI Eigen split (Geiger et al., 2012; Eigen et al., 2014) and SUN RGB-D test set (Song et al., 2015), we present the accuracy of the predicted depth map when replacing the V-layer with a single convolutional layer. As shown in Tab.(11), our V-layer can improve the depth map accuracy on both KITTI and SUN RGB-D.

Table 11: **Benefit of V-layer.** We replace the proposed V-layer with a single convolutional layer, and evaluate the predicted depth map accuracy.

| Dataset | Layer | SILog ↓ | Abs Rel ↓ | RMS↓ | RMS log↓ | $\delta_1$ ↑ | $\delta_2$ ↑ |
|---|---|---|---|---|---|---|---|
| KITTI | Convolution | 6.996 | 0.052 | 0.217 | 0.079 | 0.975 | **0.997** |
|  | **V-Layer** | **6.817** | **0.050** | **0.209** | **0.076** | **0.977** | **0.997** |
| SUN | Convolution | 13.172 | 0.098 | 0.309 | 0.132 | 0.924 | 0.981 |
|  | **V-Layer** | **12.596** | **0.094** | **0.299** | **0.127** | **0.929** | **0.983** |

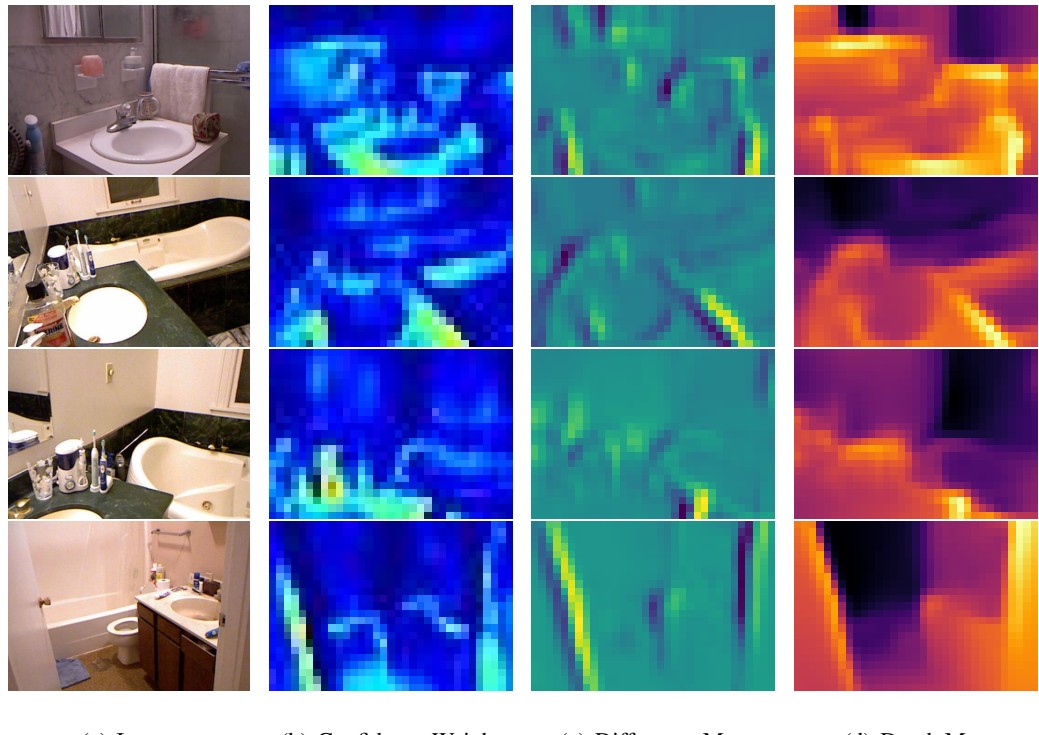

|          (a) Image          |    (b) Confidence Weight    |    (c) Difference Map    |    (d) Depth Map    |

Figure 5: **Visualization of V-layer Prediction.** We visualize the confidence weight $\Sigma_x$, the difference map $\Gamma_x$ and the depth map $Z_u$ from the V-layer when predicting on NYU Depth V2 test set.

Table 12: Comparison with AdaBins and NeWCRFs on NYU test set. All methods are trained on KITTI Eigen train set without fine-tuning on NYU.

| Method | Backbone | SILog ↓ | Abs Rel ↓ | RMS↓ | RMS log↓ | $\delta_1$ ↑ | $\delta_2$ ↑ |
|---|---|---|---|---|---|---|---|
| AdaBins(Bhat et al., 2021) | EffNet-B5+ViT-mini | 28.147 | 0.251 | 0.753 | 0.286 | 0.614 | 0.867 |
| NeWCRFs (Yuan et al., 2022) | Swin-L | 21.138 | 0.173 | 0.551 | 0.213 | 0.755 | 0.934 |
| **Ours** | Swin-L | **18.090** | **0.148** | **0.474** | **0.182** | **0.804** | **0.955** |

## A.8 MORE GENERALIZATION EXPERIMENTS

We further evaluated the generalization performance when training on KITTI and test on NYU, or training on NYU and test on KITTI. The results are shown in Tab.(12) and Tab.(13). Our method achieves better generalization performance in both settings.

Table 13: Comparison with AdaBins and NeWCRFs on KITTI Eigen test set. All methods are trained on NYU Depth V2 train set without fine-tuning on KITTI Eigen.

| Method | Backbone | SILog ↓ | Abs Rel ↓ | RMS↓ | RMS log↓ | $\delta_1$ ↑ | $\delta_2$ ↑ |
|---|---|---|---|---|---|---|---|
| AdaBins(Bhat et al., 2021) | EffNet-B5+ViT-mini | 56.871 | 0.350 | 7.221 | 0.579 | 0.434 | 0.744 |
| NeWCRFs (Yuan et al., 2022) | Swin-L | 54.460 | 0.268 | 6.246 | 0.550 | 0.512 | 0.833 |
| **Ours** | Swin-L | **42.105** | **0.221** | **5.360** | **0.426** | **0.598** | **0.888** |

## A.9 SPARSE GROUND-TRUTH DEPTH MAP

KITTI provides sparse LiDAR measurements as the ground truth (official ground truth has been inpainted to an extent). Yet, our method could work well in such cases where sparse LiDAR measurements are known. Our algorithm computes the depth map from the gradient predicted by the network in a differentiable way. Thereby, for such a case, when we apply sparse supervision on the computed depth map (note that the loss function is used for valid measurements only), the error signal from the loss function will be back-propagated to the predicted gradient to supervise the network

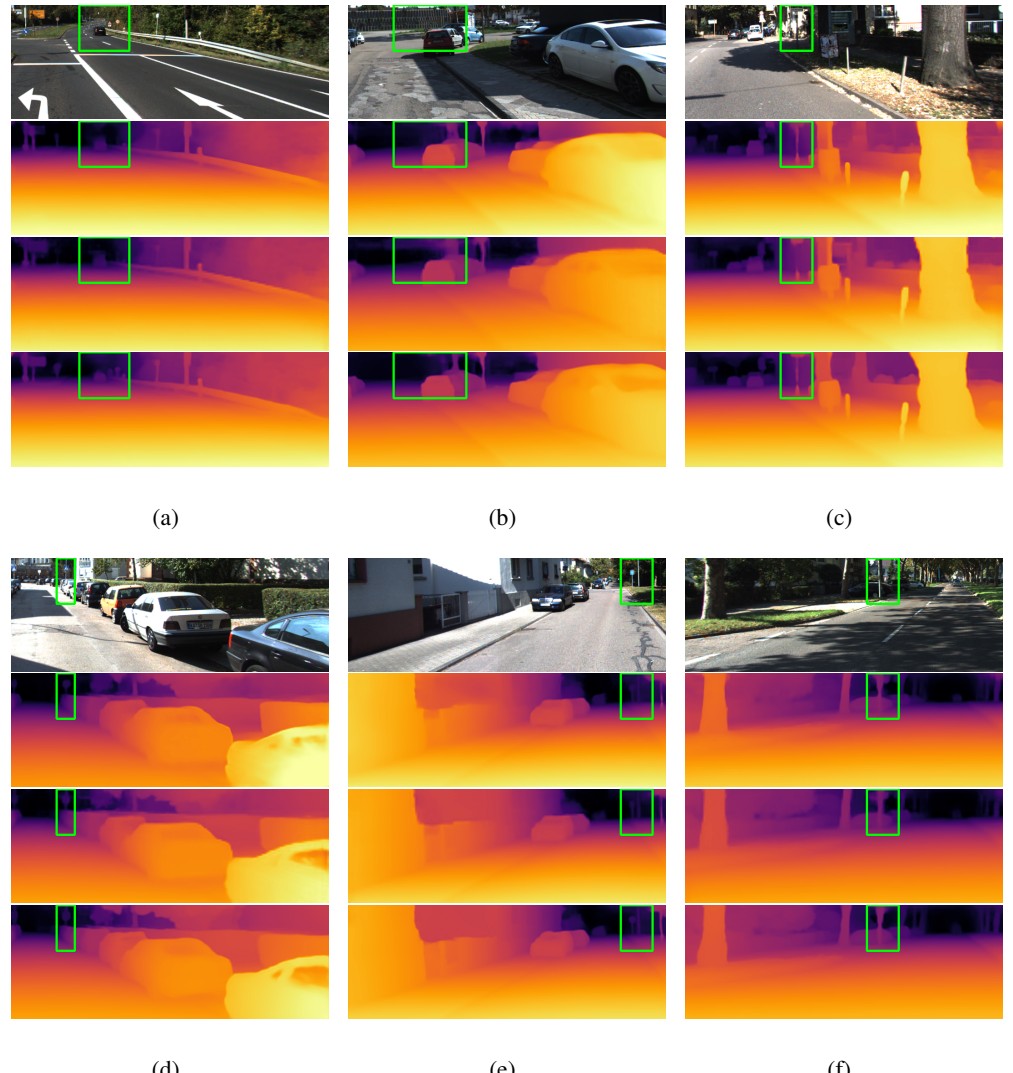

Figure 6: Qualitative comparison on KITTI Eigen split (Eigen et al., 2014). For each column, from top to bottom we present the input image, the prediction from AdaBins (Bhat et al., 2021), NeWCRFs (Yuan et al., 2022), and our framework respectively.

to learn gradient clues. We demonstrated the effectiveness of our algorithm in Tab.(2) and Tab.(3) for such cases.

## A.10  MORE VISUALIZATION

We visualize the confidence weight map $\Sigma_x$, the difference map $\Gamma_x$, and the depth map $Z_u$ from the V-layer in Fig.5. We observe that the depth value of a pixel shows correlation with respect to the image coordinates of the pixel. For example, in the last example in Fig.5, for different pixels at the door, the depth values are usually different but the first-order difference are approximately the same. This observation shows that the difference map might be easier to predict than the depth map.

## A.11  QUALITATIVE RESULTS

We provide more qualitative results on KITTI Eigen split (Eigen et al., 2014), SUN RGB-D (Song et al., 2015), and NYU Depth V2 (Silberman et al., 2012) in Fig.6, Fig.7 and Fig.8, respectively. Our framework predicts more accurate shapes and preserves the high-frequency scene information.

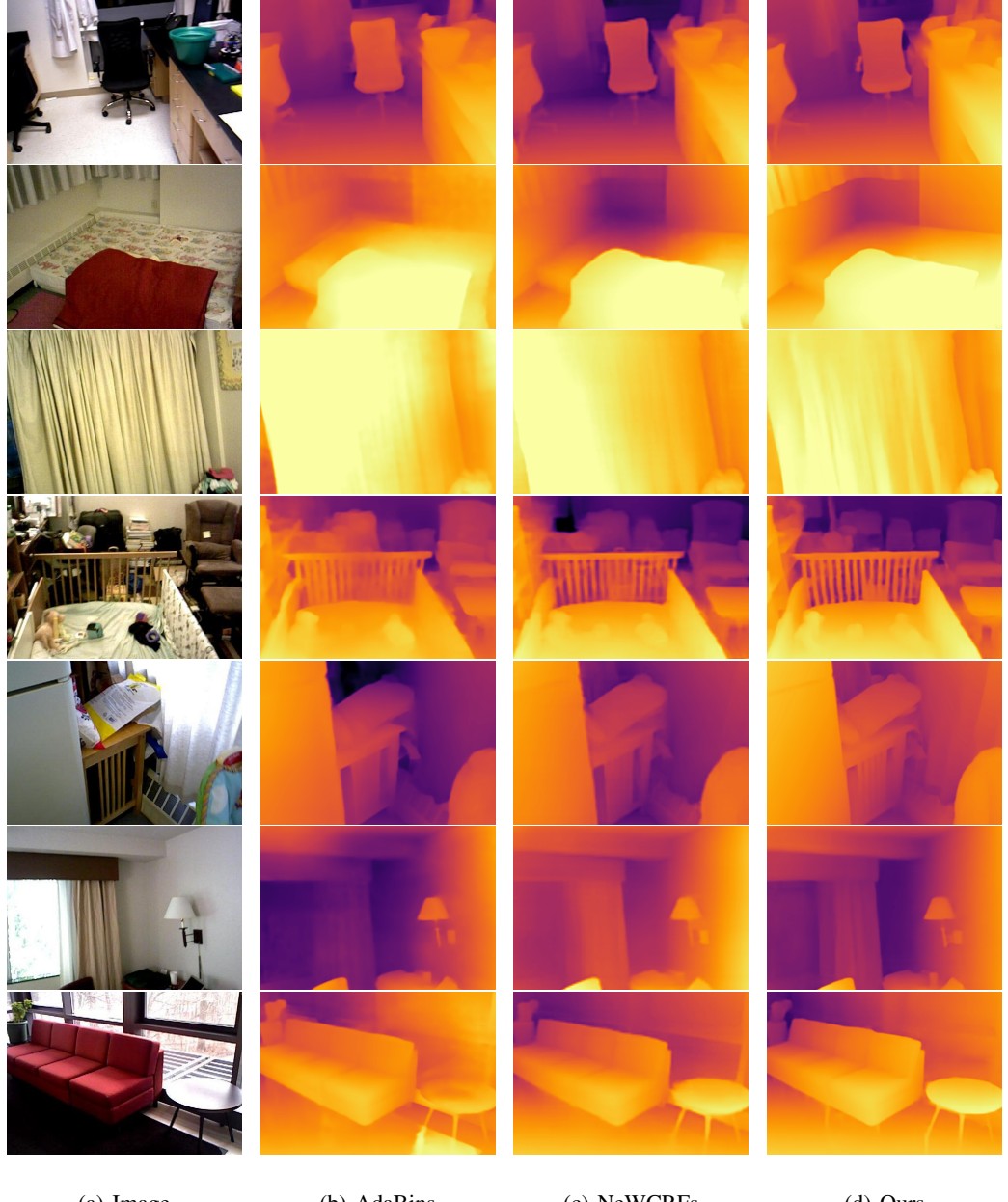

|  (a) Image | (b) AdaBins | (c) NeWCRFs | (d) Ours |

Figure 7: Qualitative comparison with AdaBins (Bhat et al., 2021), NeWCRFs (Yuan et al., 2022) on SUN RGB-D test set (Song et al., 2015). All the models are pre-trained on NYU Depth V2 (Silberman et al., 2012) training set.

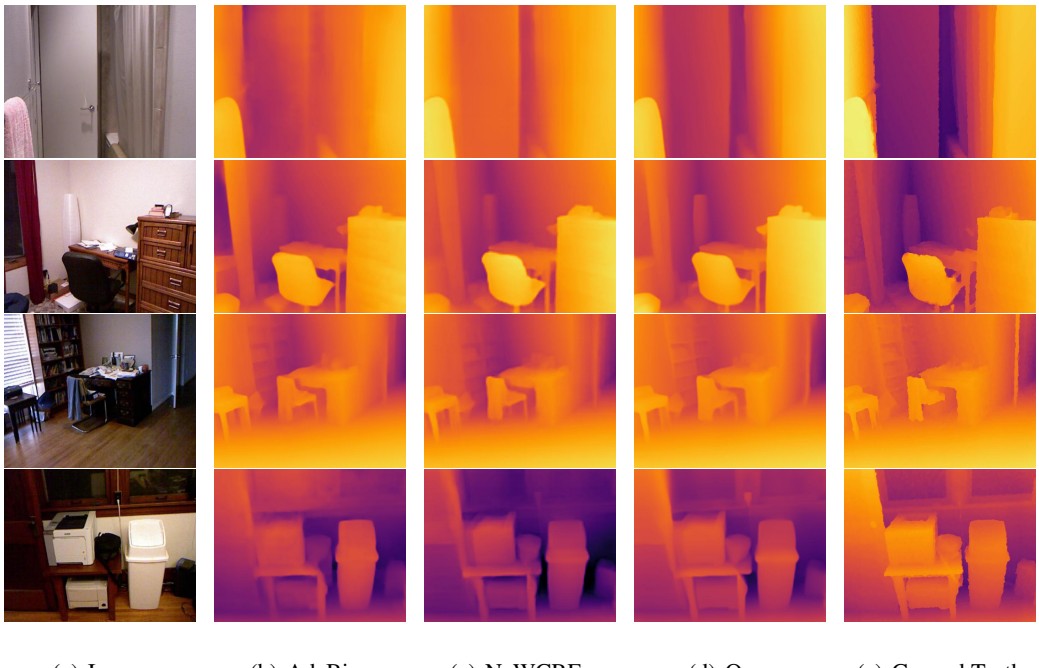

(a) Image      (b) AdaBins      (c) NeWCRFs      (d) Ours      (e) Ground Truth

Figure 8: Qualitative comparison with AdaBins (Bhat et al., 2021), NeWCRFs (Yuan et al., 2022) on NYU Depth V2 test set (Silberman et al., 2012). The ground-truth depth map are in-painted for visualization.

