# OpenReview forum: "VA-DepthNet: A Variational Approach to Single Image Depth Prediction"
_ICLR.cc/2023/Conference — ICLR 2023 notable top 25%_

### Official Review · Reviewer_PKfN · 2022-10-21

**Confidence:** 4
**Correctness:** 2
**Technical Novelty And Significance:** 3
**Empirical Novelty And Significance:** 3
**Recommendation:** 5

**Clarity, Quality, Novelty And Reproducibility:**

This paper provides the details of the network structure and the training details, so the reproducibility is high. The technical contribution is somewhat new as the variational constraint is not widely utilized in the single image depth estimation. But the design intuition is not strongly convincing.

**Strength And Weaknesses:**

1. Strength
- The proposed method outperforms very recent state-of-the-art methods in various datasets. It also achieves 2nd place in the KITTI depth benchmark.
- The dense ablation study shows the effectiveness of the V-layer and the refinement step.

2. Weakness
- It is not clearly described why the variational constraints via first-order difference are helpful for accurate depth estimation. The introduction section is more likely to be a related work section, which should describe why the proposed method (variational constraints) is required to improve the quality of the depth map.
- Figure 5 shows the evaluation of different backbones. Both Swin-S (w/o) and ConvNeXt-S (w/o) achieve 9.5, but the graph shows the error of ConvNext-S is higher than Swin-S.
- In table 1,2, it would be good to report the number of parameters or the backbone network (ResNet, Swin-S, Swin-L, etc) because the accuracy of the depth map highly depends on the backbone network. The number of parameters is reported in table 8, but the numbers are slightly different from the numbers in Table 1.
- This paper argues that the proposed method generalizes well to unseen datasets. And, the evaluations of the NYU-trained model on the SUN3D datasets are provided. The scene properties of NYU and SUN3D datasets are similar, so the evaluations on more challenging variations should be provided such as the KITTI-trained model tested on NYU, etc.

**Summary Of The Paper:**

This paper proposes the single image depth estimation method with variational constraints via first-order difference. This paper proposes V-layer which computes the depth difference map and the weight map, which will be utilized to obtain the initial depth map. The initial depth map will be upsampled and refined through the upsampling and refinement module. Finally, the metric scale depth is computed by the metric layer.

**Summary Of The Review:**

Overall, the proposed method achieves better performance than the conventional method. The dense ablation study demonstrates the effectiveness of the proposed method. However, for me, it is hard to get the design intuition of the proposed method, especially why the variational constraints via first-order difference bring performance improvement. I hope this part is clearly described in the rebuttal.

---

> ### Author Response · Authors · 2022-11-18
> **We thank you for the constructive feedback and comments. Through our response, we hope to address all your comments and concerns.**
>
> ## Q1: Significance of Variational Constraint.
>
> A1:
> An image of a general scene---indoor or outdoor, has a lot of spatial regularity. And therefore, introducing a variational constraint provides a convenient way to ensure spatial regularity (smoothness) and preserve information related to the scene discontinuities such as edges. By assuming the weighted first-order difference variation (as mentioned in our introduction last paragraph), the neighborhood depth-variation information is propagated to the network for better regularization. Following your comment, we have added a few lines in the paper’s introduction clarifying the same. We thank you for this constructive comment.
>
> ## Q2:  Figure 5 plot visualization inconsistency.
>
> A2:
> Thank you for the careful check on our plots. It was a minor slip due to the one-digit float point approximation of numerical values while plotting our figure. The SILog error for the Swin-S and ConvNeXt-S are 9.459 and 9.519, respectively. Due to the default plotting file setup, both of them become 9.5. We have updated our plot in the revised draft with exact numerical values.
>
>
> ## Q3:  In Table 1,2, it would be good to report the number of parameters or the backbone network.
>
> A3:
> We have added the information related to the backbone in Table 1 and Table 2 of our revised manuscript. We have also added more details regarding our network design in Appendix (A.4) of our revised draft for better understanding.
>
> ## Q4: The numbers in Table 8 are slightly different from the numbers in Table 1.
>
> A4:
> Since AdaBins (Bhat et al. 2021) and NeWCRFs (Yuan et al. 2022) both use the ensemble trick during evaluation, i.e., flip the input image and average the depth prediction. Such a trick slightly reduces the SILog error but significantly increases the inference time. Therefore, for a balanced comparison, we removed their ensemble trick at the test time for unbiased evaluation, resulting in a slight increase in SILog error. Following your comment, we have modified a few lines in Sec. 4.3 of our revised draft for a clear exposition of the statistics.
>
> ## Q5: Generalization
> A5:
> Following your constructive suggestion, we evaluated our method on the following settings.
> * **(a)** Training on  KITTI  and evaluating on NYU Depth V2
> * **(b)** Traning on NYU and evaluating on KITTI.
>
> In the Table Below, we report the SILog error comparison.
>
> | Method  | (a) KITTI->NYU | (b) NYU->KITTI |
> |---------|----------------|----------------|
> | AdaBins | 28.147         | 56.871         |
> | NeWCRFs | 21.138         | 54.460         |
> | Ours    | **18.090**     | **42.105**     |
>
> The above statistics indicate that our proposed approach generalizes better than the existing state-of-the-art SIDP methods. We have added these experimental results in Appendix (A.8) of our revised draft.

---

### Official Review · Reviewer_K2Ky · 2022-10-24

**Confidence:** 5
**Correctness:** 4
**Technical Novelty And Significance:** 3
**Empirical Novelty And Significance:** 3
**Recommendation:** 6

**Clarity, Quality, Novelty And Reproducibility:**

This paper has good clarity and quality. Novelty and reproducibility need improvement.

**Strength And Weaknesses:**

This article is well-written and easy to understand. The algorithm's approach is interesting and plausible. Given that only a single image is given, the prediction of depth is a very ambiguous process, so it makes sense to use more solid clues (depth gradients in this paper). The process of restoring depth by building an overdetermined system is novel.

Despite the above advantages, the novelty of this paper should be re-examined. In the last paragraph of page 2 of the paper, the authors claimed that the proposed algorithm is the first attempt to predict the first-order difference of the scene. However, a prior work (1) also predicts the first-order difference and uses them to reconstruct depth, similar to this paper's motivation. Also, (2) exploits more diverse depth derivatives for depth prediction. Related work and contributions should be revised.

(1) Monocular depth estimation using relative depth maps, CVPR 2019

(2) Multi-Loss Rebalancing Algorithm for Monocular Depth Estimation, ECCV 2020


Many depth capture devices (e.g., Lidar) can only capture depth of sparse points. On the other hand, gradient is a clue that can be used if we know the depths of all adjacent points. I think this aspect can limit the usefulness of the algorithm. Do the authors have a way around these problems?

The V-layer seems to work at a much smaller resolution (probably 1/16 or 1/32) instead of the original size resolution. However, this method results in depth prediction using gradients between regions farther away instead of two adjacent pixels. This may be a disadvantage for predicting high-frequency detail. Do the authors have any insight or empirical results regarding the resolution of the v-layer?

A more detailed description of the proposed network structure is needed for reproducibility. It would be nice if detailed layer composition was provided (even in appendix format).

---
After reading the reviews of other PCs and answers from authors,

I raise my primary score to 6.
The authors' answers alleviated most of my concerns.
Notably, Q3 and A3 will interest many researchers in this field. Including it in the paper or appendix would be nice.
Thanks to the authors for their hard work.

---

**Summary Of The Paper:**

In this paper, the authors propose an algorithm for single-image depth prediction. The key contribution of the algorithm is the V-layer, which utilizes the depth gradient to make a more accurate depth prediction. It takes the gradients in the two-axis directions and their reliability as input and obtains the depth in a linear algebraic way. The resulting low-resolution depth becomes the final depth after several upsamples and refinements. Experimental results show that the proposed method shows SOTA performance and that V-layer contributes to the performance improvement.

**Summary Of The Review:**

Overall, I think this paper very interesting. This paper is technically valid and has novel parts, but the parts I mentioned in the shortcomings should be revise. I think this paper is on the borderline.

---

> ### Author Response · Authors · 2022-11-18
> **We thank you for the constructive feedback and comments. Through our response, we hope to address all your comments and concerns:**
>
> ## Q1: In the last paragraph of page 2 of the paper, the authors claimed that the proposed algorithm is the first attempt to predict the first-order difference of the scene.
>
> A1:
> Following this comment, we re-read the entire introduction, and to our knowledge, we have not claimed our work is “the first attempt to predict the first-order difference of the scene" on page 2 of the paper. Still, it is likely that our writing could have given such an impression may be due to the repeated use of the word “first”. And therefore, we have further refined our introduction so that there is no confusion for the reader. Again, we thank you for this comment.
>
> ## Q2: Related work should be revised: Discussion on (1) and (2).
> A2:
> We have added the details related to (1) and (2) in the prior work section of our revised draft.
>
> ## Q3: Many depth capture devices (e.g., Lidar) can only capture depth of sparse points. On the other hand, gradient is a clue that can be used if we know the depths of all adjacent points. I think this aspect can limit the usefulness of the algorithm. Do the authors have a way around these problems?
>
> A3:
> Thank you for an excellent question about our approach's use for real-world applications. Fortunately, the KITTI dataset indeed relates to the comment made by you. KITTI does provide sparse LiDAR measurements as the ground truth (official ground truth has been inpainted to an extent). Yet, our method could work well in such cases where sparse LiDAR measurements are known.  Our algorithm computes the depth map from the gradient predicted by the network in a differentiable way. Thereby, for such a case, when we apply sparse supervision on the computed depth map (kindly refer to our statement in the previous draft near Eq.(7), “Note that the above loss is used for valid measurements only”), the error signal from the loss function will be back-propagated to the predicted gradient to supervise the network to learn gradient clues. We demonstrated the effectiveness of our algorithm in Tab.2 and Tab.3 for such cases.
>
>
> ## Q4:  The V-layer seems to work at a much smaller resolution (probably 1/16 or 1/32) instead of the original size resolution. However, this method results in depth prediction using gradients between regions farther away instead of two adjacent pixels. This may be a disadvantage for predicting high-frequency detail. Do the authors have any insight or empirical results regarding the resolution of the V-layer?
>
> A4:
> Indeed an insightful comment. Following your suggestion, we performed an empirical study and observed that changing the feature map resolution from 1/16 to 1/ 8 does not significantly improve the depth prediction accuracy but instead greatly reduces the FPS. Of course, your insight aligns well with our initial idea on this work. However, by conducting the experiments, we found that 1/16 is attractive both ways, i.e., computationally and in depth prediction accuracy. Below, we provide an empirical study on the same validation of choice made by us for this work. We have added this detail in the Appendix (A.6) of our revised draft.
>
> | Resolution | SIlog Error | FPS      |
> |------------|-------------|----------|
> | 1/16       | 8.198       | 7.03     |
> | 1/8        | **8.149**   | **2.59** |
>
> ## Q5: A more detailed description of the proposed network structure is needed for reproducibility.  It would be nice if detailed layer composition was provided (even in appendix format).
>
> A5:
> Thank you for the constructive suggestion. We have added extensive details related to our network structure in Appendix (A.4) of our revised draft.

---

### Official Review · Reviewer_qKm4 · 2022-10-28

**Confidence:** 4
**Correctness:** 4
**Technical Novelty And Significance:** 4
**Empirical Novelty And Significance:** 4
**Recommendation:** 8

**Clarity, Quality, Novelty And Reproducibility:**

It would be better to have a table to list the detailed framework structure and parameters in the appendix.

**Strength And Weaknesses:**

This work lays a solid contribution towards the field of single image depth prediction by introducing variational first-order constraints, showing state-of-the-art performance on mainstream public datasets. The idea is novel and introducing variational priors can be useful to a number of areas, e.g. depth prediction, scene reconstruction.

**Summary Of The Paper:**

This work proposes VA-DepthNet to solve single image depth prediction problem by exploiting classical first-order variational constraints. The proposed network disentangles the absolute scale from the metric depth and models unscaled depth map as the optimal solution to the pixel-level depth gradiant. The network focuses on the first order differences of the scene rather than pixel-wise metric depth learning. It improves the performance of depth learning in a large margin.

**Summary Of The Review:**

This work lays a solid contribution towards the field of single image depth prediction.

---

> ### Author Response · Authors · 2022-11-18
> **We thank you for the constructive feedback. We hope to address your concern and suggestion below.**
>
> ## Q1: It would be better to have a table to list the detailed framework structure and parameters in the appendix.
>
> A1:
> Following your comment, we have added the information related to the network backbone in Table 1 and Table 2.
> Furthermore, we added our network details and information regarding our hyper-parameters in the appendix (A.4)
> of our revised draft.

---

### Official Review · Reviewer_3Fva · 2022-10-29

**Confidence:** 5
**Correctness:** 3
**Technical Novelty And Significance:** 3
**Empirical Novelty And Significance:** 4
**Recommendation:** 8

**Clarity, Quality, Novelty And Reproducibility:**

Clarity is good except for equation. I don't have comments on quality, as it's good. Novelty issues are noted in the last box about [A/B/C]. Reproducibility seems good as a code release is promised.

**Strength And Weaknesses:**

Strengths:
+ Well-written and well-organized.
+ Clear new state-of-the-art results on an important problem.

Weaknesses:
- My first major criticism is that the method should be compared with other methods that implements the same idea. In Table.5, we can see that the V-layer out-performs convolution and self-attention. This is not enough. The method should be compared with [A][B][C] and demonstrate why V-layer works and the former formulations do not.
[A] Predicting sharp and accurate occlusion boundaries in monocular depth estimation using displacement fields, CVPR 2020
[B] Depth estimation via affinity learned with convolutional spatial propagation network, ECCV 2018
[C] A Two-Streamed Network for Estimating Fine-Scaled Depth Maps From Single RGB Images, ICCV 2017
- The motivation and impact of the Conv layer in Equation.8 is not clear. Although I understand that in end-to-end deep models we can always some learnable layers that mysteriously improves performance, this one needs a clearer justification because it belongs to the core technical module.
- Minor: Claiming MIDAS uses external is unfair as it is evaluated in a zero-shot setting. This can be misleading for future papers.

**Summary Of The Paper:**

This paper proposes a new network for single view depth estimation. The core technical part is a V-layer that predicts depth gradients and feature weights and solves a refined depth map. Finally, the refined depth is further refined (by black-box operators) to a larger resolution and re-scaled/shifted to the final prediction. Experimental results on NYUd2, KITTI (official and eigen splits) and SUNRGBD set a clear new state-of-the-art. Codes are promised.

**Summary Of The Review:**

Generally, I think the paper should be accepted but still has issues (see weaknesses) to be addressed. I am now voting a 6 but I can vote a 8 if convinced.

---

> ### Author Response · Authors · 2022-11-18
> **We thank you for the constructive feedback and comments. Through our response, we hope to address all your comments and concerns:**
>
> ## Q1: Comparison to the papers in the same spirit. In particular [A][B][C].
>
> A1: Thank you for this excellent suggestion.
>
> Firstly, [A][B] is generally motivated towards depth map refinement coming from an off-shelf state-of-the-art (SOTA) depth prediction network. Contrary to our proposed closed-form solution, [A][B] comprises an iterative approach.
>
> Secondly, [B] proposes using an affinity matrix to learn the relation between each pixel's depth value and its neighbors' depth values. However, the affinity matrix has no explicit supervision (or prior). It's quite possible that such a choice of learning affinity may lead to imprecise learning of neighboring relations leading to inferior results. On the contrary, our approach is mindful of imposing the first-order difference constraint leading to better performance.
>
> Thirdly, in [C], two strategies for SIDP are proposed, i.e., fusion in an end-to-end network and fusion via optimization. The end-to-end strategy fuses the gradient and the depth map via convolution layers without any constrain on convolution weights, which may not be an apt choice for a depth regression problem such as SIDP. On the other hand, the fusion via optimization strategy is based on a non-differentiable strategy, leading to a non-end-to-end network loss function. Contrary to that, our method is well-constrained and performs quite well with a loss function that helps end-to-end learning of our proposed network.
>
> Following your comment, we have added a paragraph in section 2 (Prior Work) clarifying the same. To show the effectiveness of our approach, we conducted the following experiment on the NYU Depth V2 dataset. We have added these experimental results in Appendix (A.5) for completeness.
>
> Experiments results on NYU Depth V2 dataset.
>
> | Method             | SILog     |
> |--------------------|-----------|
> | [A]                | 8.655     |
> | [B]                | 8.640     |
> | [C] (end-to-end)   | 8.557     |
> | [C] (optimization) | 8.574     |
> | Ours               | **8.198** |
>
> ## Q2: The motivation and impact of the Conv layer in Equation.8 is not clear.
>
> A2:
> Thank you for raising this concern. Such a choice is made because we use V-layer on low-resolution feature maps; as a result, we predict $S$ depth maps to fix the loss due to resolution. Here, each depth map is expected to capture different details about the scene. Therefore, we utilize the convolution layer to fuse the $S$ depth maps into a single depth map and, at the same time, compute the horizontal and vertical gradient of the depth map. We have added these details in our revised draft for further clarification and a better understanding of the readers of our choice.
>
> ## Q3: Minor: Claiming MiDAS uses external is unfair as it is evaluated in a zero-shot setting.
>
> A3:
> Thank you for the insightful comment. This minor slip is probably due to our overlooking of the GitHub code repository (https://github.com/isl-org/MiDaS) provided by the authors of MiDAS, which, in fact, has the same repo for [1][2]. [1] Vision Transformers for Dense Prediction, ICCV 2021. [2] Towards Robust Monocular Depth Estimation: Mixing Datasets for Zero-Shot Cross-Dataset Transfer, TPAMI 2022. We have corrected the corresponding notation in our revised draft.

---

> > ### Comment · Reviewer_3Fva · 2022-11-18
> > **Response**
> >
> > I would like to thank authors for not only providing a clear discussion about prior works [A][B][C], but also inlucluding convincing quantitative results. This greatly enhances the scholarship of this manuscript, which prevents us from re-inventing old methods again and again.
> >
> > I have inreased the recommendation to clear accept. I guess this is a paper that I will heavily cite in the following years, if reproducible. Hope the authors could release the code ASAP.

---

### Decision · Program_Chairs · 2023-01-20

**Decision:**

Accept: notable-top-25%

**Justification For Why Not Higher Score:**

The scope of the paper is not big enough for an oral.

**Justification For Why Not Lower Score:**

The average score may seem to be not high enough mostly due to a reviewer who gave 5 and didn't respond to the author's feedback (the issues were successfully addressed in my opinion). The contribution is solid and deserves to be known in the CV community.

**Metareview: Summary, Strengths And Weaknesses:**

The paper proposed VA-DepthNet to focus on first order differences of scene depth rather than pixel-wise metric depth by integrating classical first-order variational constraints into the encoder-decoder network. The core technical invention is a V-layer that predicts depth gradients and feature weights to solve a refined depth map. Experimental results on standard benchmarks showed substantial improvements.

In general, the reviewers are positive about the paper. R #3Fva and R# qKm4 really liked the paper, especially for obtaining the top performing results. Although R #3Fva initially had the concern that comparison with existing work on using natural image prior for deep learning is missing, R #3Fva was content with the author’s response where new experiments on NYU depth v2 dataset were reported with significant improvement. R# K2Ky raised the score to be slightly positive after the authors addressed the issues of over claim and clarified the sparse depth capturing and the low-resolution of the V-layer. R #PkfN was slightly negative about the paper with some technical issues. The authors addressed them in the rebuttal but R #PkfN didn’t respond to the authors’ new results nor to AC’s request.

The AC feels that the paper has made a good contribution to the CV community, especially the variational layer to integrate the variational constraints to a network.


**Note From Pc:**

if the above contains the word "oral" or "spotlight" please see: "oral" presentation means -> notable-top-5% and "spotlight" means -> notable-top-25%. As stated in our emails, we are disassociating presentation type from AC recommendations